# A Unified Model of Age-Related Cardiovascular Disease

**DOI:** 10.3390/biology11121768

**Published:** 2022-12-06

**Authors:** Michael Fossel, Joe Bean, Nina Khera, Mikhail G. Kolonin

**Affiliations:** 1Telocyte, Grand Rapids, MI 49503, USA; 2University of Missouri School of Medicine, Kansas City, MO 65211, USA; 3Buckingham Browne and Nichols School, Wellesley, MA 02138, USA; 4University of Texas Health Science Center at Houston, Houston, TX 77030, USA

**Keywords:** cell senescence, aging, cardiovascular disease, telomere, telomerase, gene therapy, senolytics, autophagy, senotherapeutics

## Abstract

**Simple Summary:**

Age-related cardiovascular disease is the foremost cause of death globally. Despite medical advances, we have been unable to stop or reverse the underlying pathology. We offer a model that is consistent with all clinical and laboratory data, can explain the clinical heterogeneity of age-related disease both within (e.g., human patients) and between species, and is predictively valid for interventional clinical trials. Cell aging is not only the central mechanism of such disease, but studies of human cells, of human tissues, and in animals demonstrate that we can effectively reverse cell aging at the cellular, genetic, and epigenetic levels. Given current biotechnology techniques, the reversal of cell aging is now technically feasible in human trials. The implications for our ability to cure and prevent cardiovascular disease are without historical precedent. This approach may not only allow us to intervene effectively in age-related cardiovascular disease but will likely lower (rather than raise) the costs of global medicine.

**Abstract:**

Despite progress in biomedical technologies, cardiovascular disease remains the main cause of mortality. This is at least in part because current clinical interventions do not adequately take into account aging as a driver and are hence aimed at suboptimal targets. To achieve progress, consideration needs to be given to the role of cell aging in disease pathogenesis. We propose a model unifying the fundamental processes underlying most age-associated cardiovascular pathologies. According to this model, cell aging, leading to cell senescence, is responsible for tissue changes leading to age-related cardiovascular disease. This process, occurring due to telomerase inactivation and telomere attrition, affects all components of the cardiovascular system, including cardiomyocytes, vascular endothelial cells, smooth muscle cells, cardiac fibroblasts, and immune cells. The unified model offers insights into the relationship between upstream risk factors and downstream clinical outcomes and explains why interventions aimed at either of these components have limited success. Potential therapeutic approaches are considered based on this model. Because telomerase activity can prevent and reverse cell senescence, telomerase gene therapy is discussed as a promising intervention. Telomerase gene therapy and similar systems interventions based on the unified model are expected to be transformational in cardiovascular medicine.

While aging is the defining element and the greatest predictive risk factor for age-related cardiovascular disease [1,2], it is generally ignored in favor of other risk factors, such as genetic, environmental, and clinically defined factors in discussions of both causes and interventions. The impact of age-related cardiovascular disease is dramatic. It constitutes the most significant and costliest set of health problems in most developed nations. Even those who do not die of cardiovascular disease often die with cardiovascular disease. In 2019, at least 48% of Americans had some form of cardiovascular disease [3], a percentage rising with age [4]. As a comorbidity, it contributes to other debilitating conditions, such as chronic kidney disease, dementia [5], etc. Cardiovascular diseases diminish the quality and length of life and prove expensive personally, nationally, and globally [6], both directly and indirectly. In 2010, the global cost was $863 billion; this cost is estimated to reach $1.044 trillion by 2030 [6,7].

## 1. Current Approaches to Cardiovascular Disease Intervention

Although cardiovascular disease research and its implementation may be at least partially responsible for an observed 70% reduction in mortality, efficacy has plateaued, with mortality decreasing only 1% per year since 2011 (and actually increasing by 1% in 2015). The NIH annual budget is more than $2.3 billion for cardiovascular disease, with more than $350 million for stroke research alone [8,9]. The emphasis has proven beneficial: 44% of the decrease in mortality between 1980 and 2000 has been attributed to modifiable risk factors [10,11], yet aging per se is commonly ignored. A list of cardiovascular disease research areas typically fails to mention aging [12], instead focusing on other factors [13,14,15], including lifestyle risks (although fewer than 5% of patients institute changes) [16], detection (e.g., even where interventions are unavailable), medications (e.g., ACE inhibitors, ARBs, PCSK9 inhibitors, etc.) [17], surgical approaches, interventional cardiology (e.g., transcatheter valve repair, the use of forearm arteries for CABG, improvements in thrombectomy, etc.) [18,19], and genetic risks (e.g., cholesterol and lipid metabolism [20], etc., even when not amenable to intervention) [21].

The primary clinical approach to cardiovascular disease prevention has been to manage risk factors [22]. The American College of Cardiology and the American Heart Association highlight exercise, obesity, type-2 diabetes, cholesterol/dyslipidemia, hypertension, tobacco use, and aspirin use as having an impact on cardiovascular disease risk [23]. Such factors may affect the risk for both strokes and myocardial infarctions [24,25] and healthy lifestyles may decrease cardiovascular disease risk [26,27], yet the approach remains insufficient, with increasing obesity, decreasing physical activity, and a plateau in risk improvement over the past decade [28].

Age per se [29] is regarded as a non-modifiable risk factor, paralleling the response to known genetic factors, including family history [30,31], sex, ethnicity, etc. The cardiovascular disease prevalence in the US rises from ~40% in 40–59-year-olds, to ~75% in 60–79-year-olds, and 86% in those above 80 [4]. Above 40 years of age, there is a near doubling of cardiovascular disease per decade adjusting for other risk factors [32]. Above 55 years of age, the risk of stroke doubles per decade [33,34]. The key variable is not chronological, but biological age, a more appropriate indicator for risk assessment [35,36,37], treatment, and theoretical modeling, leading to a search for more accurate measures of biological aging, with mixed results [38,39,40,41,42,43,44,45]. Beyond “upstream” risk factors or “downstream” biomarkers, a nuanced understanding of cell aging may offer a more comprehensive model and an optimal point of intervention, viewing cell aging as a modifiable factor.

Current pharmaceutical approaches offer clinical efficacy comparable to lifestyle modifications [46], although they may be costly, with efficacy limited to specific populations or syndromes. Moreover, pharmaceuticals may be seen as targeting secondary aspects of cardiovascular disease, rather than fundamental processes that underline the pathology. Current surgical approaches have specific indications, high costs and risks, and the assessment of optimal interventions remains in flux [47]. We can treat anginal symptoms without necessarily addressing anginal pathology. We can intervene in coronary artery disease without altering the ongoing, parallel pathology in other untreated arteries. This concern is equally apt for all current surgical, medical, and lifestyle interventions: they demonstrate significant clinical value, without necessarily altering the fundamental, cell- and genetic-level disease processes that underlie age-related cardiovascular disease. Quoting Thomas Sydenham (1624–1689), “A man is only as old as his arteries”. We suggest that this needs to be updated: we are only as old as the cells in those same arteries. In summary, current surgical and pharmaceutical approaches target secondary outcomes rather than fundamental processes. They do not sufficiently or effectively reverse the fundamental pathology of cardiovascular disease.

## 2. Future Approaches to Cardiovascular Disease Intervention

Despite extensive information on risk factors and pathological outcomes, we lack a unified systems model of cardiovascular dysfunction resulting from aging. Extant models of cardiovascular disease are “component” rather than systems models, viewing disease as the result of downstream biomarkers (e.g., plaque, foam cells, intercellular endothelial cell gaps, a loss of tensile strength or elasticity, fibrosis, etc.) rather than demonstrating the age-related mechanisms taking us from upstream risk factors to downstream pathology and clinical outcomes. The absence of a unified systems model may thwart our ability to effectively prevent and cure age-related cardiovascular disease. While having a significant impact on mortality and morbidity, current interventions do not prevent or cure disease, remain expensive, and may have significant associated risks. 

Interventions aimed at the cellular origin of the pathology would have a better chance of preventing and treating cardiovascular disease, lowering healthcare costs, and incurring less risk. The development of optimally effective clinical interventions requires a more comprehensive and accurate unified model of aging-associated cardiovascular disease. This model should account for: (i) the role of aging per se, (ii) how “upstream” risk factors result in “downstream” clinical findings, (iii) the heterogeneity of clinical findings, timing, and comorbidities, as well as interspecies heterogeneity. The model should suggest novel, feasible, and effective points of intervention. Such a model should be “unified” in explaining the heterogeneity between patient presentations (e.g., stroke, CAD, cardiac myopathy, aneurysms, etc.), as well as the heterogeneity within patient presentations (e.g., age of onset, progression, etc.), and the heterogeneity between species (e.g., humans, mice, non-human primates, etc. The model should be a “systems” model in that it offers an explanation for the effects of genetic, epigenetic, and behavioral differences on downstream clinical and histologic findings. Specifically, the model should take us from the multiplicity of risk factors (age, infection, trauma, hypertension, hyperglycemia, diet, radiation damage, toxic exposures, etc.) to the gamut of clinical and histologic outcomes of cardiovascular disease. 

The inclusion of cell aging as a central process suggests a point of intervention that may prove more effective than current therapies [48,49,50,51]. Cell aging was first described more than sixty years ago [52], and taking it into consideration should have important medical and therapeutic implications [53,54]. As Hayflick succinctly observed: “The cause of aging is ignored by the same people who argue that aging is the greatest risk factor for their favorite disease [50,55]”. Clinically, age-related disease can be viewed as the collective result of cell aging, the stage upon which all age-related pathology occurs [56].

The continuum of cell aging (Figure 1A) is characterized by a gradual decline in cell function, while cell senescence is the endpoint of this process. Cell senescence is characterized by an irreversible cell-cycle arrest, senescence-associated gene expression (Figure 1B), and pro-inflammatory senescence-associated secretory phenotype (SASP) [57]. The alteration of gene expression is not an all-or-nothing event, occurring only in the terminal phase of cell aging, but rather occurs as a progressive alteration. Cell aging occurs when telomerase (an enzyme coded for by the *TERT* gene and required for telomere maintenance) is inactivated, allowing these protective chromosomal caps to shorten with age [58]. Upon telomerase inactivation, telomere shortening proceeds along with other poorly understood global epigenetic changes in gene expression (which may be non-canonical) [59]. Thus, pathology begins to develop as a result of cell aging long before cells ever become fully senescent.

Reduced telomerase activity and critical telomere shortening lead to replicative cell senescence as the result of cell aging. A plethora of variables—including oxidative damage, DNA damage response, inflammation, tumor suppressor genes (e.g., p53, p21, p16), cell cycle checkpoint regulation (e.g., CHK1, CHK2), cytokine signaling, etc.—are associated with the cell senescence phenotype. In fact, the expression of SA-beta-galactosidase (Figure 1B), INK4a (p16), and of other senescence markers and SASP may occur without TERT inactivation or telomere attrition. In addition, upstream genetic, epigenetic, and lifestyle factors—including trauma, infection, hyperglycemia, hypertension, tobacco, alcohol, etc.—may accelerate cell division, the rate of telomere shortening, and cell aging generally. While the detailed mechanisms of the regulatory processes are incompletely understood, the overall effect has been extensively demonstrated, as have the effects on tissue. Senescent cells (i) are incapable (in replicative senescence) of replacing lost cells; (ii) trigger inflammation, thus damaging surrounding tissue (SASP); and (iii) have altered gene expression that contributes to cell dysfunction. 

### 2.1. The Role of Cell Aging in Age-Related Cardiovascular Disease

Cell aging has been theorized [53,54,56,60] and implicated in age-related disease in animals [61,62] and humans [50,63]. The unified systems model (Figure 2) suggests that cell aging mediates the cascade of events between upstream clinical risk factors and downstream clinical outcomes. Traumatic injury may accelerate cell aging in joints, UV exposure may accelerate cell aging in the skin, and having two APOE4 alleles may accelerate microaggregate formation, even in the context of relatively mild microglial cell aging, resulting in earlier and more severe Alzheimer’s disease. The downstream heterogeneity of clinical outcomes (specific diagnosis, age of onset, course of disease, individual clinical findings, etc.) results from the upstream heterogeneity of genetic, epigenetic, and behavioral history. However, pathology plays out in the unified “stage” of cell aging (Figure 2).

Cell aging is typified by a deceleration in molecular turnover, whether intracellular, extracellular, or intranuclear, and whether protein, lipid, or other molecular moieties, resulting in a gradual increase in the percentage of dysfunctional molecules. Slower molecular turnover results in an increased *proportion* of damage, but the *rate* of damage can also increase as a result of genetic or environmental effects acting synergistically with cell aging. Upstream risk factors can not only increase the *rate* of cell aging [64,65,66,67] but can also exacerbate the *effects* of cell aging. For example, rheologic trauma (e.g., at bifurcations, the aortic arch, etc.) and barotrauma (e.g., hypertension) may increase the rate of cell aging and thereby hasten the onset of pathology (e.g., aortic aneurysm), but genetic risks are also operative [68,69,70,71].

Cardiovascular disease is the result of vascular endothelial cell aging [72,73], as well as aging in smooth muscle cells [74,75], cardiomyocytes [76], cardiac fibroblasts [77], and immune cells [78]. In aging endothelial cells, there is a decreased endothelial nitric oxide synthase (eNOS) and nitric oxide, critical in atherogenesis and hypertension. Endothelial cell aging increases monocyte adhesion, implicated in atherogenesis, and the induction of endothelial cell aging results in typical atherogenic changes. Importantly, these processes are linked with telomerase activity. In the case of laminar fluid stress, telomerase is required for normal endothelial cell response [79], while telomere shortening is associated with increasing arterial stiffness [80]. This vascular dysfunction is ameliorated or normalized by telomerase, causing reversion to a pattern of gene expression typical of younger endothelial cells [81,82], indicting endothelial cell aging in atherogenesis [83]. Upstream risk factors leading to cell aging and downstream outcomes predisposing to cardiovascular disease are important to integrate into the Unified Model.

### 2.2. Upstream Risk Factors in Cardiovascular Disease

Upstream risk factors are clinical starting points that operate within the context of cell aging and cause age-related disease. Pathology can result from normal cell aging, from accelerated cell aging (e.g., hypertension, diabetes, vascular infections, tobacco use, poor diet, etc.), or from accelerated molecular damage (e.g., genetically abnormal free radical scavengers, membrane lipids, mitochondrial enzymes, etc.). Upstream risks may be categorized as genetic, epigenetic, or behavioral.

Genetic risks may be relatively asymptomatic in early life, as in patients with aberrant elastin or collagen genes [84], but increasingly symptomatic in later life as molecular pools are recycled far more slowly and the percentage of damaged molecules rises. Risks may be genetic, as with abnormal cytochrome C oxidase or mitochondrial genes [85,86], but interactive processes may also occur, for example in endothelial cell aging [67,87,88] and aortic smooth muscle cells [89], where oxidative damage [66] and cell aging work in tandem [90]. Epigenetic risks may include regulatory elements (which greatly outnumber protein-expressing genes), including variability in inherited telomere lengths [91]. Epigenetic differences manifest as regulatory idiosyncrasies: patients with identical genes may have different patterns of expression and resulting differences in age-related disease.

Telomere length varies among newborns, while it is similar in different organs of the human fetus [92]. Genetic mechanisms involved in the regulation of *TERT* expression and telomerase activity are described in recent reviews [93]. Germline mutations in *TERT* and other genes that control chromosome termini result in the shortening of telomeres transmitted to the progeny. This leads to progressive telomere length changes over successive generations. Somatic mutations in telomerase machinery, as well as the epigenetic downregulation of TERT and its co-activators, account for tissue-specific differences in telomere attrition and senescence onset during aging. Signaling pathways including WNT, converging on KLF4, as well as c-MYC, GA-binding proteins (GABP), and other E26 transformation-specific (ETS) family transcription factors have been found to regulate *TERT* transcription [94]. Epigenetic mechanisms also include promoter methylation, histone deacetylase inhibitors, and miRNAs [95]. The post-translational regulation of TERT activity is also important to consider [96]. The age of telomerase inactivation predetermines the rate of telomere attrition, which varies among individuals [91]. In combination with somatic genetic and epigenetic changes in the oxidative damage protection pathways, these variables in turn predetermine senescence onset in distinct cell populations and the resulting aging of individual tissues.

Behavioral risks (e.g., smoking [97,98,99], air pollution [11,100,101,102], diet [103], alcohol, etc.) may either increase the *rate* of cell aging or exacerbate the *effects* of cell aging. An example from the central nervous system may clarify. Head trauma or CNS infections would increase the risk of Alzheimer’s disease by increasing the *rate* of cell aging, while a patient with no history of trauma or infection but who is biallelic for APOE4 is prone to earlier microaggregate formation, exacerbating the *effects* of cell aging. There is a growing body of evidence that the rate of telomere shortening is also a function of environment and behavior. 

Cell aging is universal, while the risk, timing, and severity of the resulting age-related cardiovascular diseases are defined by genetic, epigenetic, and behavioral starting points. Specific downstream clinical outcomes—myocardial infarction, stroke, varicosities, cardiomyopathy, etc.—depend on the specific upstream starting points.

### 2.3. Downstream Outcomes in Cardiovascular Disease

Downstream outcomes—biomarkers, clinical syndromes, or pathology—vary because upstream risk factors vary, even in the face of a unified fundamental process, that of cell aging. In atherosclerosis, for example, the initial formation of a fatty streak, composed of lipid-laden macrophages (foam cells) within the intima, is followed by vascular smooth muscle cell proliferation, fibrous plaques, cholesterol deposits, and inflammatory cell recruitment [104]. The observable fatty streak formation is, however, preceded by a less obvious inflammatory response within the endothelium, which has been linked to a number of insults, including hypertension, diabetes, tobacco use, and the inflammatory secretory pattern directly associated with cell aging [105]. This results in the recruitment of inflammatory cells (e.g., monocytes) and the loss of adhesion in endothelial cell junctions. The failure of cell junction integrity causes the leakage of plasma components, including LDL cholesterol, into the subendothelial compartment. Inflammation further contributes to endothelial damage and predisposes to LDL phagocytosis. Recruited monocytes transform into macrophages, phagocytosing oxidized LDL and forming atherosclerotic foam cells [104]. They also secrete inflammatory factors, induce smooth muscle cell proliferation and often form fibrotic tissue. 

The earliest fatty streaks in children and adolescents lack this fibrous component and other pathological features that characterize atherosclerosis in elderly patients where cell aging is prominent [105,106,107]. Pathology can occur without cell senescence, for example in the presence of significant physical or oxidative damage to the endothelium, as evidenced by arterial fatty streaks in many patients by age twenty. However, the arterial remodeling typical of the young [105,106,107,108] is slower or absent as cell aging progresses and other changes, such as mitochondrial dysfunction, occur [76,109,110]. Cell aging drives the progression of atherosclerosis even without classical risk factors [111,112] and is critical to the progression of significant pathology [107,108]. Early lesions may regress with effective clinical intervention in upstream risk factors, but advanced lesions, in which cell aging is prominent, are not [106,111,112,113]. Even aggressive intervention in risk factors (e.g., diabetes, hypertension, and cholesterol) are insufficient, as the cell aging of the arterial cells previously induced by such factors continues apace [78,83,104,106,111,114,115]. The vascular endothelial cell aging continues, and pathology accelerates [78,83,111,114,115].

Hutchinson-Gilford progeria provides an example of the role of cell aging and telomere shortening in atherosclerosis. In these children, their lamin-A mutation results in telomere lengths typical of 70-year-old patients. Although lacking classical risk factors—diabetes, tobacco use, hypertension, or elevated cholesterol—they develop atherosclerotic pathology [116] and usually die (average lifespan 12.7 years) of atherosclerotic disease [117,118]. Mouse models of H g progeria likewise show an impairment of mechano-signaling in endothelial cells, suggesting that cell aging and their resulting senescence contributes to excessive fibrosis and cardiovascular disease [119], and studies on human progeric cells suggest that endothelial-targeted therapy may be effective [120]. Cardiovascular disease occurs in the context of accelerated cell aging, even when other risk factors are absent.

Cardiovascular risk factors such as diabetes, hypertension, obesity, and dyslipidemia may be promoted by cell aging and senescence, but may also accelerate cell aging, resulting in a vicious cycle of pathology. Ongoing damage accelerates cell aging, leading to subendothelial exposure, persistent and irreversible inflammation, the progression of atherosclerotic plaque, reduced fibrous plaque coverage, and eventual rupture [78,83,114,115]. This is all the more serious with LDL elevation, as high cholesterol leads to increased plaque and the oxidation of LDL within the intima, with increased inflammatory responses [106,111,112,113].

A similar causal cascade occurs in aneurysms, as the cell aging of the vascular walls results in structural deterioration. The vascular mesenchymal stromal cells (VMSCs) from aortic aneurysms are more senescent than those from healthy aortas, and medial smooth muscle cells taken from aneurysms are likewise more senescent than cells from normal arteries in the same individual [121,122,123].

Cardiac failure is also linked to cell aging. Cardiomyocytes [48,124] and cardiac fibroblasts [77] display the hallmarks of cell aging, including dysfunctional DNA repair, mitochondrial dysfunction, impaired contractility, and increased fibrosis, with the subsequent impairment of ejection fraction and heart failure [76,124,125,126]. Curiously, experimentally induced transient [127] or premature senescence [128] may reduce perivascular fibrosis and inflammation, perhaps by converting actively dysfunctional (though aged) cells into far less active, replicative senescent cells. Age-related increases in arrhythmias may also be linked to cell aging [129].

### 2.4. Heterogeneity of Clinical Presentations in Cardiovascular Disease

Age-related cardiovascular disease is clinically heterogenous, encompassing a spectrum of diseases that share a temporal predilection (aging) and a tissue of origin (the cardiovascular system) but often little else, at least superficially. There may appear to be little in common between strokes, myocardial infarctions, varicose veins, and capillary pruning. However, such diseases all demonstrate vascular endothelial cell aging and secondary organ pathology [130].

In the case of CNS arterial disease, both hemorrhagic and thrombotic stroke can result from the cell aging of the vascular endothelial cells, the former from arterial rupture and the latter from arterial blockage. In the case of coronary disease, myocardial infarction results from insufficient flow secondary to vascular endothelial cell aging. Other presentations typical of aging patients [131]—varicose veins, chronic venous insufficiency, deep vein thrombosis, capillary pruning, etc.—can be the result of vascular endothelial cell aging, with the consequent loss of integrity and structural resilience, whether in arterial, venous, or capillary walls.

Clinical heterogeneity is common. Patients of identical age may vary in disease, vascular localization, onset, or course, and this heterogeneity likely results from the heterogeneity of the initial risk factors. Individuals differ genetically, epigenetically, and behaviorally. Genetic differences may include alleles which impact cholesterol metabolism, inflammatory processes, immune responses, the efficacy of hepatic detoxification, renal physiology, collagen strength, elastin production, or a myriad of other genetic “starting points” prior to the changes incurred by cell aging. Epigenetic differences may be equally profound: rather than an abnormal gene, there is an abnormal pattern of gene expression, with subtly different responses to physiologic stimuli. Behavioral history—alcohol use, diet, exercise, infectious disease, trauma, etc.—also generates clinical heterogeneity in cardiovascular disease.

Genetic risk factors have become increasingly well-established over the past decade with the increased use of genome-wide association studies (GWAS) to identify disease-susceptibility loci [132,133,134]. However, epigenetic diagnostic markers and therapeutic targets, independent of genetic inheritance, have also been identified [135,136,137]. While associations between the risk factors and the clinical outcomes are commonly acknowledged, the cascade of events proceeding via cell aging, presenting as overt changes in tissue and organ function, and ultimately as clinical disease, are less well appreciated.

Contributing to the confusion, many studies analyze aging in cells that are not responsible for the disease [138,139]. Because of sample collection simplicity, circulating leukocytes rather than vascular endothelial cells are commonly analyzed for telomere attrition [140,141,142,143,144,145,146,147,148,149]. As might be expected, there is a rough correlation in telomere lengths between different tissues since the entire organism is undergoing progressive cell aging over time. However, the same might be said of the correlation between cardiovascular disease, CNS neurodegenerative disease, osteoarthritis, osteoporosis, and other age-related diseases as the patient ages. The correlation reflects the age of the patient but is not sufficiently specific to the tissue in question. Although too often measured in patients with cholesterol abnormalities [150] or cardiovascular disease [151,152,153,154,155,156], leukocyte telomere lengths can change for independent reasons, such as infections or autoimmune reactions. Relying upon leukocyte measures in assessing cardiovascular disease results in unwarranted conclusions based on inappropriate correlations and irrelevant cell choices. In assessing risk factors for age-related cardiovascular disease, such as diabetes [157,158] or vascular aging [159] we see the same error in measuring telomeres in leukocytes [160,161] rather than in relevant cells. Because vascular dysfunction in cardiovascular disease is largely attributable to endothelial cells [90] and/or vascular smooth muscle cells [75], telomere lengths in such cells should be used as a gauge of cell aging and atherosclerosis [162]. Studies that focus on the measurement of telomeres and cell aging of the appropriate cells find a clearer picture [163,164,165]. The data show (and reviews have emphasized) that telomere shortening and cell aging are not only implicated in age-related cardiovascular aging but precede and are more prominent in affected tissues [166,167]. These studies underline the role of cell aging in multiple age-related cardiovascular diseases [168], including hypertension [169,170], and atherosclerosis [83,171,172], predisposing to heart failure. 

Identifying cell aging as a fundamental mechanism underlying cardiovascular disease offers a more complex—but perhaps a more accurate—paradigm of disease etiology. Current diagnostic approaches to cardiovascular disease test for either upstream risk factors or downstream biomarkers. High cholesterol [150,173] or hypertension, for example, are clearly upstream risk factors but may operate through the acceleration of cell aging [90,174,175,176]. Coronary calcium scans [177], on the other hand, are a downstream biomarker, but cell aging may be the driving force behind plaque buildup [178].

Upstream risk factors such as chronological age, genetic risk, diet, stress, and trauma contribute to cell aging, while downstream biomarkers of cardiovascular disease, such as venous changes, arterial wall hardening, and myocardial infarctions are the clinical consequences of cell aging. Changes in endothelial physiology, including the eNOS/NO and FGF21 pathways, as well as calcium signaling, appear to be detrimental [87]. Other metabolic changes, including hypertensive or hyperglycemic injury to these cells [173], are also important [179]. Cardiovascular cell aging can also cause secondary CNS disease [180], whether as an acute insult (e.g., stroke) or as a chronic result of vascular insufficiency and changes in the blood–brain barrier. Indeed, telomere shortening and cell aging are prominent not only in Alzheimer’s but also in vascular dementia and other age-related neurodegenerative diseases [181,182,183].

There is accumulating evidence that vascular endothelium aging underlies age-related cardiovascular pathology [63,178]. Endothelial aging induced by experimental telomere attrition stimulates metabolic disorders through SASP [110,184]. Healthy angiogenesis is not an active process in most mature organs, yet endothelial cell division and capillary sprouting underlie normal maintenance in muscle, adipose tissue, and likely the brain. Muscle regeneration relies on angiogenesis, and dysfunctional angiogenesis is implicated in some cardiovascular diseases. Endothelial cell aging is also likely to underlie peripheral artery disease (PAD) and capillary pruning. Although cause-and-effect may be complicated [185,186,187,188], cell aging offers a novel interventional target in both PAD and type-2 diabetes (T2D) [189,190]. While most cardiac diseases may be attributable to vascular endothelial cell aging, other cell types are also implicated. Cardiomyopathy may be the result of cardiac fibroblast aging [191]. Pericytes, mural cells [192], macrophages [193], and other myeloid cells (with foam cell and plaque formation) may also be involved [194]. Mice lacking *TERT* in perivascular progenitor cells [195] have premature metabolic disease [196], more so if given a high-caloric diet [197].

Adipose tissue, a key reservoir of senescent cells, may indirectly contribute to the aging of cardiovascular cells [198,199,200,201]. In obesity, the death of hypertrophic and hypoxic adipocytes prompts leukocyte infiltration, inflammation, fibrosis, and reduced ability to contain triglycerides, which results in steatosis and may contribute to the development of T2D [202]. Early in life, adipocyte progenitor cells divide continuously [203,204], replacing dysfunctional hypertrophic adipocytes [205,206,207]. With age, adipocyte progenitors lose replicative capacity, which is accelerated by hyperlipidemia [150,208,209], obesity [210,211,212,213], and oxidative stress [214,215]. Telomere length in progenitor cells defines the ability of adipose tissue to expand by hyperplasia, rather than hypertrophy [197], perhaps explaining the lack of T2D in “healthy obese” individuals [196]. In mice lacking *TERT*, adipocyte progenitors undergo premature replicative senescence, and T2D development is accelerated [195]. Obesity development also involves increased endothelial cell division (as well as leukocyte infiltration) to vascularize expanding adipose tissue [196]. The accelerated cell aging may aggravate adipose dysfunction, the accompanying dyslipidemia [216], and hypertension [217], independently of the cardiovascular risk factors [218]. All of these aging cells secrete SASP factors with systemic vascular effects. Aging monocytes from adipose tissue may also disseminate systemically and may contribute to atherosclerosis and inflammation directly. Potential long-distance effects of other types of aging cells in adipose tissue remain to be investigated [219].

## 3. Perspective on Optimal Clinical Intervention

Current clinical interventions in age-related cardiovascular disease are inadequate. A more complete understanding of the cell aging process may indicate innovative points of intervention.

### 3.1. Component Interventions in Cardiovascular Disease

Broadly, we may divide interventions into those targeting components of age-related disease and those targeting systems aspects of aging. Component interventions target upstream risk factors (e.g., hypertension, smoking, etc.) or downstream biomarkers (e.g., cholesterol levels [220,221,222,223,224], myocardial fibrosis [225,226,227], macrophage recruitment [228], arterial blockage, etc.) [229]. Systems interventions target aging as a fundamental cellular process.

Component interventions, while useful, have limited clinical efficacy and neither stop nor reverse the age-related pathology. Component interventions include dietary or lifestyle interventions, small-molecular pharmaceuticals to control hypertension, hyperglycemia, or cholesterol levels, surgical interventions to improve coronary artery flow or treat aneurysms, etc. Such interventions may lower risk [230], slow the disease course, or be emergently life-saving, yet have no effect on the underlying pathology and cell aging. For example, vascular stents, coronary artery bypass, or angioplasty, targeting downstream findings, may improve flow in diseased arteries without improving function in the senescent vascular endothelial cells lining those same arteries, however. Such procedures do not affect the underlying disease process. Component approaches targeting upstream risk factors have benefits, but neither stop nor reverse the decline in cell function. Dietary and exercise interventions, for example, lower the risk of recurrent cardiovascular events [231], perhaps slowing the rate of cell aging in adipose tissue [200], muscle [232], or vasculature [233,234], yet the underlying progress of cell aging remains.

Recently, senolytic therapies, aiming to remove senescent cells, have been touted [49,73], as have senomorphics [235] (small molecules mainly aimed at SASP), including their use as an intervention in cardiovascular disease [51,219,236]. The problem is that senolytics are expected to prompt additional division in the remaining cells in order to replace cells removed by senolytic therapy. This engages a vicious cycle of cell senescence and potentially exacerbates the pathology that senolytics were intended to treat. As a result, transient improvement may be followed by more rapid pathological decline than would have occurred without senolytic intervention [237]. Senomorphics are aimed to modulate the senescent cells, which is likely to be a more viable rejuvenation approach. However, current senomorphics target specific pathways and are neither all-inclusive of downstream effects, nor do they address the upstream and more fundamental causes of SASP (Figure 3).

Some cardiovascular drugs—statins, beta blockers, clopidogrel, warfarin, ACE inhibitors, and aspirin—may alter the pace of cell aging (for example, through reducing inflammation, which is linked to cell aging) [238]. Beta-blockers may protect endothelial cells from the negative effects of aging [239]. ACE inhibitors, traditionally used to block the effects of angiotensin, may reduce the adverse effects of cell aging [240]. Finally, aspirin may actually reduce endothelial cell aging [241]. None of these drugs have proven sufficiently effective as interventions in age-related cardiovascular disease. New, systemic approaches may prove far more effective.

### 3.2. Systems Interventions in Cardiovascular Disease

Suggestions for effective intervention in age-related disease [242] by targeting cell aging were first proposed two decades ago [53,54]. The cell aging model offers a consistent rationale for the effects of diverse upstream risk factors, for the expression of those risk factors in downstream biomarkers, and for the heterogeneity of clinical disease. This model views cell aging and its explicit components of *TERT* inactivation, telomere shortening, and altered patterns of gene expression (potentially due to a telomere-independent function of *TERT*) [243,244] as the major driver of age-related disease (Figure 3). The model suggests that other changes commonly accompanying cell aging, such as inflammatory changes, increased oxidative damage, metabolic abnormalities, and DNA damage, are largely the results, rather than the causes, of cell aging. While each of these (and other dysfunctional changes) contribute to cell aging, creating a vicious cycle of damage, they are in large part due to the more fundamental cellular genetic and epigenetic changes that underlie cell aging and lead to cell senescence.

The slower turnover of molecular pools, characteristic of aging cells, plays a critical role. Mitochondrial enzymes, for example (99% of which are nuclear in origin), turn over more slowly, yielding a decline in the ATP/ROS ratio. The mitochondrial membrane lipids also turn over more slowly as cell aging progresses. The turnover rate of scavenger molecules (e.g., catalase, SODs, etc.) also decelerates, so ROS production increases, leakage increases, and scavenging fails. Nuclear membranes are also turned over more slowly, hence the increased risk of DNA damage in the face of a decline in DNA repair efficiency. Oxidative damage may therefore be the result of cell aging, as well as potentially contributing to cell aging [56]. Recent studies provide evidence that some of these processes can be reversed by inducing *TERT* expression [82]. That cell aging increases mitochondrial dysfunction and oxidative damage provides a reasonable explanation for the lack of cumulative damage found in germ cells and inherited mitochondria.

The Unified Model offers a novel point of intervention: using telomerase to target cell aging, rather than targeting cell aging biomarkers [191]. Intervention in cell aging, the crux of the age-related disease process, is feasible using a variety of potential approaches that share the optimal target of adjusting telomere lengths directly [245].

Preventing and reversing cell aging and age-related changes in cells, tissues, and animals includes in vitro work in human cells [246], in vitro and ex vivo work in human tissues [81,247,248], and in vivo work in animals [61,249]. The promise of this approach is supported by data from various approaches inducing telomerase activity [62,250,251,252,253,254]. These include the use of *TERT*-activating compounds in acute coronary syndrome [255], whether via canonical or non-canonical effects [256]. A potentially promising telomerase activator is danazol, an antiestrogenic/anti-progestogenic hormone. Danazol was shown to suppress telomere attrition in patients with telomeropathies, the congenital diseases of premature telomere loss, which predispose to bone marrow failure, liver cirrhosis, and pulmonary fibrosis [257]. Another small molecule reported to activate *TERT* is TA-65 (cyclastragenol), isolated from *Astragalus* genus plants [70]. TELO-SCOPE, a national, multi-center, double-blind, placebo-controlled, randomized trial, is underway to assess the effect of danazol-induced telomere extension in pulmonary fibrosis. Telomerase activation by peroxisome proliferator-activated receptor agonists, such as pioglitazone, has also been reported [258].

Gene therapy is a direct approach to telomerase reactivation in aging somatic cells. Vectors (including AAV [259,260,261], anellovirus [262], lentivirus [82], mRNA [263], or lipid nanoparticles [264]) can deliver the human telomerase gene (*hTERT*) effectively. AAV vectors have already proven effective in treating Spinal Muscular Atrophy (SMA) [265]. Because the vascular endothelium is directly accessible, systemic agents to reset vascular cell aging may be highly effective. Telomerase gene therapy has been reported to delay aging in mouse models [262]. Consistent with that, gene therapy with the gene *TRF1*, a telomerase co-factor, prolongs mouse health spans [266]. Patients with progeroid diseases, which predispose to expedited aging, are likely to greatly benefit from telomere extension therapy [94]. In mouse models of Hutchinson-Gilford progeria, TERT gene therapy was shown to reverse vascular senescence and extend the lifespan. In translational pursuits, the same group showed that the transient delivery of mRNA encoding *TERT* rapidly extends telomeres in human cells [267] and suppresses the hallmarks of progeroid cells [268]. In a recent study, intranasal and injectable gene therapy has shown promise for healthy life extension in animal models [269]. Telomerase activation is yet to be clinically tested as a therapeutic approach to CVD prevention and treatment. However, there are reports that exercise and the use of statins, the two proven approaches to CVD prevention, reactivate telomerase activity [258]. There is much hope that telomerase gene therapy will provide new approaches to treating CVD [246]. In a preclinical mouse model, improved ventricular function and the reduction in infarct scars after acute myocardial infarction was achieved through *TERT* gene therapy [270]. While telomerase gene therapy has not been approved by the US Food and Drug Administration, clinical trials have begun. Libella Gene Therapeutics is evaluating the safety and tolerability of AAV-*TERT* currently with the goal of treating critical limb ischemia, Alzheimer’s disease, and aging. Telomere Therapeutics is another company that has initiated analogous trials in Spain to target pulmonary fibrosis and renal fibrosis as aging-associated diseases.

While various concerns have been voiced regarding *TERT* therapy [271], one issue that has been repeatedly raised is that of the possible risk of cancer induction. However, TERT gene therapy in mice resets cell aging and increases longevity without increasing the frequency of cancers [262,272]. Not only is the actual relationship a remarkably complex one [273] but a growing body of data suggests that the prevention of cell aging by telomerase may protect against cancer [242]. For example, it has been reported that the CRISPR-mediated correction of the *TERT* promoter prolongs the survival of mice with experimental cancers [274]. While clinical trials will determine the actual risk of *TERT* gene therapy, it appears to be misrepresented for cancer-free recipients [275]. Even patients with cancers overexpressing telomerase are unlikely to be jeopardized by *TERT* reactivation in benign cells, although they are possibly an increased risk group. Our understanding of potential *TERT* vector immunogenicity, non-target tissue effects, dosing, and route issues remains in flux. However, the clinical potential of telomerase activation to suppress the development of or treat aging-associated diseases such as CVD is tantalizing, and the results of clinical trials are highly anticipated.

## 4. Conclusions

While cardiovascular diseases are multifaceted and complex, the discussed evidence suggests that cell aging may be the fundamental driver for age-related cardiovascular diseases. The most promising therapeutic intervention for age-related cardiovascular disease is cell rejuvenation. Interventions targeting cell aging have the potential to far exceed the efficacy of current interventions by altering the fundamental processes that underlie and drive age-related cardiovascular disease. The approach proposed and discussed here is not to *remove* but to *reset* aging cells by using telomerase gene therapy. The reversal of gene expression to that characteristic of younger cells promises to be more effective and longer-lasting than other current therapies. The clinical outcome may involve not merely slowing age-related cardiovascular disease but the reversal of the pathogenic tissue effects that result from cell aging. It is imperative to explore not merely the efficacy but the safety of this approach in regulated, rigorous, and credible human trials. Fundamental interventions at the level of cell aging to treat age-related cardiovascular disease may lower medical costs for three reasons: (i) treatment may lower the costs of chronic disease and long-term medical care; (ii) R&D (and production) costs may be amortized over a massive patient base, and; (iii) ongoing advances in efficient delivery methods are lowering current costs. In addition, there should be secondary benefits in lowering the percentage of GDP devoted to health insurance and other national medical costs, as well as the secondary effects upon personal, national, and global economies, which can result from returning an individual’s ability to participate positively in such economies.

## Figures and Tables

**Figure 1 biology-11-01768-f001:**
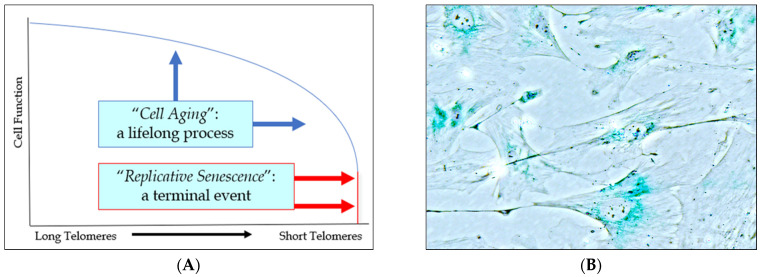
**Cell aging and senescence.** (**A**) Cell aging is a continuum characterized by a gradual decline in cell function, while cell senescence is the endpoint of this process. (**B**) Fibroblasts in culture undergoing senescence, as detected by blue SA-beta-galactosidase staining (courtesy of Zhanguo Gao).

**Figure 2 biology-11-01768-f002:**
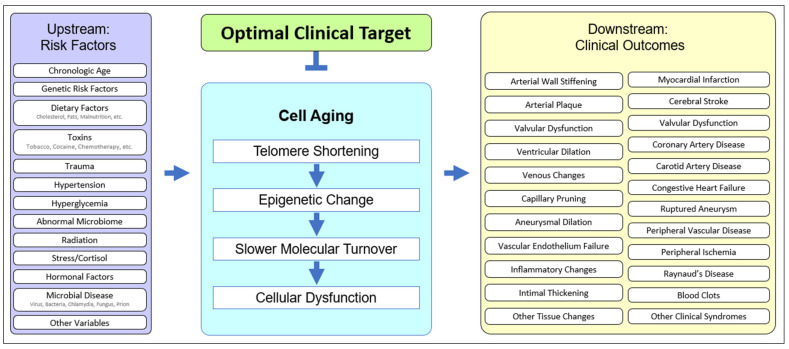
**The unified model of cell aging being central to age-related cardiovascular disease.** Upstream risk factors provide variable starting points, affect cell aging, and result in variable downstream clinical outcomes, suggesting an optimal point of intervention.

**Figure 3 biology-11-01768-f003:**
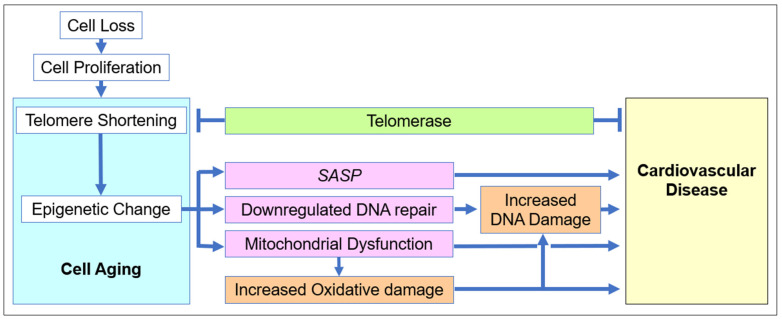
**The cascade of pathology in age-related cardiovascular disease.** During aging, cell divisions lead to telomere shortening upon *TERT* inactivation. This results in the global alteration of gene expression. Combined with the subsequent damage accumulation in DNA and other cell components, this process leads to cell senescence and SASP, which ultimately accounts for cell and tissue dysfunction. Telomerase, being a pivotal link in the process, is a potentially effective point of intervention.

## Data Availability

Not applicable.

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
