# Peer review of "A Unified Model of Age-Related Cardiovascular Disease"

_biology, 2022, doi:10.3390/biology11121768_

Round 1
Reviewer 1 Report
Authors proposed a model unifying the fundamental processes underlying most age-associated cardiovascular pathologies. According to this model, cell aging, leading to cell senescence, is responsible for the tissue changes leading to age-related cardiovascular disease.
In general, the paper is well written however However, the epigenetic and genetic aspects need to be expanded and addressed in the appropriate way. In the manuscript there is no clear explanation of these mechanisms and how they impact or direct cellular senescence.
Author Response
We thank the Referee for the constructive comments, which are now fully addressed in the revised manuscript. Epigenetic and genetic variables that may pre-dispose to age-related cellular senescence are discussed on page 5. We have now expanded this discussion on page 5 as follows:
Telomere length varies among newborns while it is similar in different organs of the human fetus (https://doi.org/10.1203/00006450-200209000-00012). Genetic mechanisms involved in the regulation of TERT expression and telomerase activity are described in recent reviews (https://doi.org/10.3390/epigenomes6010009). Germline mutations in TERT and other genes that control chromosome termini result in shortening of telomeres transmitted to progeny. This leads to progressive telomere length changes over successive generations. Somatic mutations in telomerase machinery, as well as epigenetic downregulation of TERT and its co-activators, account for tissue-specific differences in telomere attrition and senescence onset during aging. Signaling pathways including WNT, converging on KLF4, as well as c-MYC, GA-binding proteins (GABP) and other E26 transformation specific (ETS) family transcription factors have been found to regulate TERT transcription (10.1016/j.cell.2020.12.028). Epigenetic mechanisms also include promoter methylation, histone deacetylase inhibitors, and miRNAs (10.3390/cancers13061213). Post-translational regulation of TERT activity is also important to consider (https://doi.org/10.1186/s12964-019-0372-0). The age of telomerase inactivation predetermines the rate of telomere attrition, which varies among individuals (10.1016/j.ebiom.2020.103164). In combination in somatic genetic and epigenetic changes in the oxidative damage protection pathways, these variables in turn predetermine senescence onset in distinct cell populations and the resulting aging of individual tissues.
Reviewer 2 Report
In this review, Fossel et al. proposed a unified model which the cell aging is the fundamental process underlying most age-related cardiovascular pathogenesis. They summarized the upstream risk factor and downstream clinical outcomes of age-related cardiovascular diseases and proposed that cell aging the connection between these two components. They discussed potential therapeutical approaches by targeting cell aging, especially telomerase gene therapy, are promising interventions for age-related cardiovascular diseases. Generally, this review is well-written, and the proposed model is promising. It will be better to include a comprehensive discussion on the current progress, including clinical and non-clinical studies, of telomerase gene therapy on cardiovascular diseases, while the current discussion on this part looks quite brief. It will be more informative to provide the current knowledge on how the telomerase gene therapies were conducted, how it recognized the target cells, the outcomes of the therapy, as well as the side-effects. It is necessary to provide a summary table showing the current studies using telomerase gene therapies on diseases, especially cardiovascular diseases.
Author Response
We thank the Referee for the constructive comments, which are now fully addressed in the revised manuscript. We discuss the current progress in clinical and non-clinical studies aimed to re-activate telomerase on page 10. Due to the lack of telomerase gene therapy clinical trials, especially aimed at cardiovascular diseases, there is not enough material for a table. However, we have now expanded the discussion starting on pages 10-11. Specifically, we added the following paragraph (provided here without references) that addresses clinical telomerase therapy developments:
Gene therapy is a direct approach to telomerase reactivation in aging somatic cells. Vectors (including AAV, anellovirus, lentivirus, mRNA, or lipid nanoparticles) can deliver human telomerase gene (hTERT) effectively. AAV vectors have already proven effective in treating Spinal Muscular Atrophy (SMA). Because the vascular endothelium is directly accessible, systemic agents to reset vascular cell aging may be highly effective. Telomerase gene therapy has been reported to delay aging in mouse models. Consistent with that, gene therapy with the gene TRF1, a telomerase co-factor, prolongs mouse health span. Patients with progeroid diseases, which predispose to expedited aging of, are likely to greatly benefit from telomere extension therapy. In mouse models of progeria, TERT gene therapy was shown to revers vascular senescence and extend lifespan. In translational pursuits, the same group showed that transient delivery of mRNA encoding TERT rapidly extends telomeres in human cells and suppresses the hallmarks of progeroid cells. In a recent study, intranasal and injectable gene therapy has showed promise for healthy life extension in animal models. Telomerase activation is yet to be clinically tested as a therapeutic approach to CVD prevention and treatment. However, there are reports that exercise and the use of statins, the two proven approaches to CVD prevention, re-activate telomerase activity.There is much hope that telomerase gene therapy will provide new approaches to treat CVD. In a preclinical mouse model, improved ventricular function and reduction of infarct scars after acute myocardial infarction was achieved upon TERT gene therapy. While telomerase gene therapy has not been approved by the US Food and Drug Administration, clinical trials have begun. Libella Gene Therapeutics is evaluating safety and tolerability of AAV-TERT currently with the goal of treating critical limb ischemia, Alzheimer’s disease, and aging. Telomere Therapeutics is another company that has initiated analogous trials in Spain to target pulmonary fibrosis and renal fibrosis as the aging-associated diseases.
Please note that the preceding and subsequent paragraphs were also expanded to address the suggestion to elaborate on the preclinical and clinical developments.